# Early Childhood Oral Health and Nutrition in Urban and Rural Nepal

**DOI:** 10.3390/ijerph16142456

**Published:** 2019-07-10

**Authors:** Chloe Tsang, Karen Sokal-Gutierrez, Priya Patel, Brett Lewis, Debbie Huang, Kristina Ronsin, Ashmita Baral, Aparna Bhatta, Nehaa Khadka, Howard Barkan, Sidhanta Gurung

**Affiliations:** 1University of California Berkeley, Berkeley, CA 94720, USA; 2School of Dentistry and Oral Health, A.T. Still University Arizona, Mesa, AZ 85206, USA; 3School of Public Health, University of California Berkeley, Berkeley, CA 94720, USA; 4Harvard T.H. Chan School of Public Health, Boston, MA 02115, USA; 5School of Medicine, Oregon Health and Science University, Portland, OR 97239, USA; 6School of Public Health, Columbia University Mailman, New York, NY 10032, USA; 7World Health Organization, Copenhagen, 2100 Capital Region, Denmark; 8Self-Reliant Center, Kathmandu 44700, Nepal; 9School of Public Health, University of California Los Angeles Fielding, Los Angeles, CA 90095, USA; 10University of Texas McGovern Medical School, Houston, TX 77030, USA

**Keywords:** oral health, nutrition, diet, dental caries, urbanization, mothers, child, Nepal

## Abstract

Globalization and urbanization in Nepal have driven a nutritional transition from an agricultural-based diet to an ultra-processed, sugary diet. This study assessed the nutrition and oral health of 836 children age 6 months to 6 years and their families in rural and urban Nepal. Mothers were interviewed about maternal–child oral health and nutrition, and children received dental exams and height and weight measurements. Analyses utilized SPSS statistical software. Most families lived within a 5-minute walk to a store selling ultra-processed snacks and sugary drinks. While most mothers knew sweets caused tooth decay, half of the children were given sweets daily, and 58.2% of children had dental caries. Caries began in the first 2 years and increased in prevalence and severity to age 6, when 74.3% had caries and 20% experienced mouth pain. Despite greater health knowledge and resources among urban mothers, urban children’s increased access to junk food and frequency of consumption was associated with higher prevalence and severity of caries compared to rural children. Severe caries was associated with malnutrition, especially in rural children. Preventive strategies are needed in early childhood to incorporate nutrition and oral health education and dental care into maternal–child health services, and develop policies to prohibit the sale of junk food around schools.

## 1. Introduction

Over recent decades, globalization and urbanization have led to a nutritional transition in low- and middle-income countries from traditional, agricultural-based diets to ultra-processed and high-sugar diets, particularly for children [1,2,3]. This dietary shift has contributed to a double-burden of child malnutrition and obesity, increased risk for type 2 diabetes, and cardiovascular diseases [4]. It has also led to a high prevalence of early childhood caries (ECC) or tooth decay, affecting 50–90% of children worldwide [5].

ECC is caused by dietary sugars metabolized by oral bacterial strains, which produce acid and demineralize tooth surfaces, forming cavities [6]. Numerous behavioral and socioeconomic risk factors for caries have been identified, including prolonged bottle-feeding, frequent snacking and sugary beverage consumption, inadequate toothbrushing, lack of fluoride and dental care, and both low and high income [7,8,9].

Globally, limited access to dental care leaves most ECC untreated, which can lead to widespread loss of tooth structure, chronic infection and inflammation, mouth pain, decreased appetite, disrupted sleep, and poor functioning and confidence in school, all of which interfere with children’s quality of life, growth, and development [9,10]. In low- and middle-income countries, studies have documented a significant association between ECC and malnutrition; a bidirectional relationship is postulated, where undernutrition predisposes a child to caries and caries predispose a child to undernutrition [11,12,13].

Nepal is experiencing economic growth, urbanization, and a nutritional transition amidst long-standing child malnutrition associated with geographic isolation, poor infrastructure, political instability, social discrimination, and natural disasters [14]. National nutrition surveys have found that 41% of children under age 5 were stunted, 29% were underweight, and 11% were wasted, with higher rates of malnutrition in rural areas [14]. Dental studies of 5–6 year-old Nepali children have confirmed prevalence of caries ranging from 52% to 79%, with high rates of mouth pain [15,16,17,18,19]. While these studies identified multiple contributors to caries, they did not explore the differences in oral health and nutrition between rural and urban populations, nor the associations between severe caries and malnutrition.

This study aims to examine socio-behavioral risk factors for ECC in rural and urban populations and associations between caries and malnutrition in Nepali children to help guide interventions to improve children’s health and wellbeing.

## 2. Materials and Methods

This is a cross-sectional study of the oral health and nutrition of a convenience sample of young children and their mothers and caregivers from low-income urban and rural sites in Nepal. The study was conducted in collaboration with the Self-Reliant Center (SRC), a Nepali non-profit, non-governmental health organization. Ethical review and approval were provided by the University of California, Berkeley, Committee for Protection of Human Subjects (2010-06-1655), and Nepal Health Research Council (105/2011).

SRC staff recruited 5 urban communities in the Kathmandu and Lalitpur districts and 5 rural communities in the Sindhupalchok district. Local community leaders, health staff, and teachers were engaged to set up community-based health camps for data collection in 2011. All families with children aged 6 months to 6 years were invited to participate. Trained community health workers obtained mother or caregiver written, informed consent and child verbal assents when age-appropriate. They interviewed mothers or caregivers regarding maternal and child nutrition and oral health knowledge, practices, and experience, utilizing a survey instrument modified from the World Health Organization (WHO) oral health survey [20]. Licensed Nepali dentists performed child dental exams by visual inspection with a light and mirror, recorded decayed, missing, and filled teeth (dmft), and estimated depth of cavitation into the enamel, dentin, or pulp based on WHO standards [20]. The dentists calibrated their exams by independently and then jointly examining 5 children and agreeing on findings. Trained health workers and student volunteers weighed and measured the children, without shoes and in light clothing, using a stadiometer and digital scale (Seca), according to WHO standards [21]. After completing the data collection, dentists and health workers provided oral health and nutrition counseling, toothbrushes, fluoride toothpaste, fluoride varnish application, and referrals for any necessary dental treatment.

Child nutrition status was determined using WHO AnthroPlus software to calculate Z-scores for height-for-age (HAZ), body mass index (BMI)-for-age (BAZ), and weight-for-age (WAZ). Z-scores < −2 were considered malnutrition (moderate or severe), with low HAZ indicating stunting or chronic malnutrition, low BAZ indicating wasting or acute malnutrition, and low WAZ indicating underweight, or chronic or acute malnutrition [21,22]. Data analysis with SPSS (version 25) yielded descriptive statistics for the total sample as well as urban and rural subsamples, and inferential analyses for independent samples to test for statistical significance. Statistical procedures for bivariate and multivariate analyses were guided by measurement scaling of the variables and sampling (independent vs. paired sampling). Association analyses relied on the computed correlation coefficients and binomial logistic regression.

## 3. Results

### 3.1. Demographics

The study sample included 836 children and 632 mothers; approximately two-thirds from urban sites and one-third from rural sites. Overall, mothers had a mean age of 30 with 5 years of education and families had a mean of 2−3 children. Rural families had significantly lower education and less access to potable water and non-wood cooking fuel, which reflected a lower socioeconomic status (Table 1).

The study measured families’ access to non-nutritious snacks and sugary beverages, such as candy, chips, and soda (i.e., junk food), by the time it takes to walk from home to a store that sold those products. Overall, junk food was highly accessible, with approximately three-quarters of families living within a 5-minute walk to a convenience store. Urban families had significantly greater access to junk food than rural families, with nearly 9 out of 10 urban families compared to half of rural families living within a 5-minute walk to a store (Table 1).

### 3.2. Mothers’ Knowledge of Oral Health and Nutrition

To assess mothers’ knowledge of ECC, an open-ended question was asked, “What do you think causes tooth decay in children?” Nearly 9 in 10 mothers knew that eating sweets caused tooth decay, but only 1 in 4 mothers knew that not brushing children’s teeth caused decay, and few mothers knew that soda or juice and bottle-feeding contributed to ECC. Overall, knowledge of risk factors for caries was slightly higher for urban mothers than rural mothers (Table 2).

### 3.3. Mothers’ and Children’s Nutrition Practices

One-third of mothers reported drinking milk daily, with higher rates among rural mothers than urban mothers. Fewer than 3% of mothers reported daily consumption of soda or junk food.

In both urban and rural populations, nearly all children were breast-fed, for a mean of 2–3 years. In addition, 1 in 5 (22.2%) urban children were bottle-fed, nearly triple the rate of that for rural children (7.9%). Overall, children were bottle-fed to a mean of 16 months. Although most children were given only milk in the bottle, 1 in 10 were given sugary drinks in the bottle. While 1 in 4 urban children were given the bottle in bed, two-thirds of rural children were given the bottle in bed.

Two-thirds of children drank milk at least once a day. Overall, half of the children ate sweets once a day or more, with two-times higher rates in urban children compared to rural children (62.0% vs. 29.7%). Additional analysis revealed that 1 in 4 children ate sweets multiple times a day, with four-times higher rates in urban compared to rural children (37.2% vs. 9.2%). In addition, 1 in 4 children consumed other junk foods such as chips and biscuits daily, with three-times higher rates in urban compared to rural children (31.2% vs. 10.3%). Few children drank soda or juice daily. Examining children’s daily junk food consumption by age showed that from ages 0–2 years, urban and rural children had similar rates of consumption. However, from 3−6 years of age, the rate of daily junk food consumption doubled for urban children and remained stable for rural children, contributing to the urban–rural disparity in junk food consumption (Table 2). 

### 3.4. Mothers’ and Children’s Oral Health Practices

Regarding oral healthcare at home, most mothers reported that they currently owned a toothbrush. Urban children were more likely than rural children to have their own toothbrush (84.4% vs. 56.0%) and toothpaste (93.7% vs. 84.2%). Only 1 in 5 mothers reported helping their children brush, with higher rates in urban than rural families (25.6% vs. 15.5%). Regarding access to healthcare services, 8 in 10 mothers reported receiving prenatal care with an average of 5 visits, but only half of mothers reported ever visiting a dentist, with higher rates in urban than rural areas. Although nearly all children were reportedly up-to-date on their immunizations, only 1 in 10 children had visited a dentist, with urban children twice as likely than rural children to have had a dental visit (12.4% vs. 6.3%) (Table 2).

### 3.5. Children’s Oral Health Status

Overall, 58.2% of children experienced tooth decay, with a significantly higher prevalence of tooth decay in urban children compared to rural children (62.2% vs. 51.2%). Overall, 93.6% of the tooth decay was untreated. The number of decayed teeth ranged from 0 to 19 teeth, with a mean of 3 decayed teeth. The mean decayed, missing, and filled teeth (dmft) score for urban children was higher than for rural children. Among children with tooth decay, half (50.1%) had 1–4 decayed teeth, one third (35.3%) had 5–9 decayed teeth, and 1 in 7 children (14.6%) had 10 or more decayed teeth (i.e., over half of their baby teeth), with similar severity distribution in urban and rural groups. Approximately 1 in 5 children (19.6%) had deep decay into or near the pulp, and 1 in 5 children (20.0%) experienced mouth pain, with a higher prevalence of mouth pain reported in urban children compared to rural children (22.7% vs. 15%). Overall, 1 in 5 mothers reported that their child’s oral health was “bad”, compared to only 1 in 20 reporting that their child’s overall health was “bad” (Table 3).

Figure 1 shows children’s caries experience by age. Overall, a greater prevalence and severity of tooth decay is seen for urban children compared to rural children. Tooth decay is seen to begin in the first 2 years of life and progress steadily thereafter in prevalence and severity, becoming a common and severe experience by age 6. At age 6, urban children had a higher prevalence of tooth decay compared to rural children (84.1% vs. 56.5%) and greater severity of caries as evidenced by a higher mean number of decayed teeth (4.6 vs. 2.6), higher prevalence of deep decay (35.4% vs. 24.2%), and higher prevalence of mouth pain (42.2% vs. 16.1%).

### 3.6. Children’s Nutrition Status

Overall, one-third of children were moderately or severely stunted, with higher rates in rural children compared to urban children (42.7% vs. 29.5). Approximately 15% of children were underweight, and only 2% were wasted, with no significant differences between urban and rural children (Table 3).

### 3.7. Associations Between Oral Health and Nutrition

Maternal nutrition practices were correlated with child nutrition practices. Mothers with frequent consumption of milk were more likely to have children with frequent consumption of milk and mothers with frequent consumption of junk food were more likely to have children with frequent consumption of junk food. Among children, the frequency of consumption of soda or juice and sweets was correlated with their caries severity by dmft. Children’s caries severity by dmft was correlated with the likelihood of experiencing mouth pain, and with both chronic and acute malnutrition by height-for-age (HAZ), weight-for-age (WAZ), and BMI-for-age (BAZ) scores. Severe caries into the pulp also showed a significant correlation with malnutrition by HAZ and WAZ scores, particularly among the rural children. Mouth pain was correlated with chronic malnutrition by HAZ score (Table 4).

## 4. Discussion

This study examined socio-behavioral factors associated with children’s oral health and nutrition, and the relationship between children’s oral health and nutrition status in urban and rural Nepal. Junk food was readily accessible within a 5-minute walk from homes and prominent around schools, in both urban and rural areas. Most mothers knew that sweets caused caries, but they had limited knowledge of other caries risk factors, such as bottle-feeding, sugary drinks, and not brushing their children’s teeth. While mothers generally followed a traditional diet and rarely consumed junk food, they were 10-times more likely to give their children junk food daily than to consume it themselves. Although mothers knew that junk food was unhealthy, children as young as 8 months were given junk food, and most children were given junk food at least once a day. Many mothers explained that they relied on sweets to stop their children from fussing or to incentivize and reward good behavior.

Studies have shown that junk food advertising on television, billboards, front-of-store placement at children’s eye-level, and sales in and around schools encourage children to ask for, buy, and consume junk food [23,24]. In Nepal, these marketing strategies are evident, and the nutrition transition can be seen in one generation, from parents to children. Once ultra-processed and sugar-laden snacks and beverages have become a daily staple of children’s diets, there is substantial risk for adverse nutritional outcomes, including severe tooth decay, malnutrition and obesity, type 2 diabetes, and cardiovascular disease.

Improving mothers’ oral health and nutrition knowledge is an important initial step in improving children’s oral health and nutrition, but socio-environmental factors can challenge parents’ ability to put their knowledge into practice, particularly when unhealthy choices are widely-marketed, highly-accessible, convenient, low-cost, and palatable [23,24]. There is a need for more public information about good nutrition, encouragement to continue traditional diets, and healthier guidance for child behavior management. In addition, parents’ nutrition choices should be supported by policies to establish healthier food environments, such as prohibiting the sale of junk food in and around schools, implementing taxes on sugary drinks and snacks, and ensuring access to clean water [8,23,24].

Regarding oral health practices, most families reported owning toothbrushes and toothpaste, although there was some uncertainty about whether toothpaste contained fluoride. Only a minority of mothers helped their children brush, despite pediatric and dental recommendations for adults to help children brush until at least age 6 [25]. This suggests the need for public oral health promotion messages to encourage parents and teachers to help children brush their teeth more effectively, both at home and at preschool.

Examining children’s oral health status, over half of the children had untreated caries, with higher prevalence and severity in urban children compared to rural children. While some dental epidemiologists have asserted that ECC is “a disease of poverty” [26], our study and some others in developing countries have found that lower-income and rural settings may be protective against ECC by preserving traditional dietary patterns and limiting children’s access to junk food [27,28,29]. For low-income populations globally, it is a hopeful message that achieving good oral health does not require higher education or substantial income; it can be achieved by simple, healthy daily practices in nutrition and oral hygiene.

This study demonstrated the risk factors and onset of ECC beginning during the first 2 years of life and increasing in prevalence and severity throughout early childhood. Similar to previous studies, we found that a majority of 5–6 year-old Nepali children had untreated tooth decay, and 1 in 5 children suffered from deep decay and mouth pain that could interfere with their eating, sleeping, and concentration in school [14,15,16,17,18]. We also found a significant association between severe early childhood caries and both chronic and acute malnutrition, similar to other dental studies in low- and middle-income countries [10,11,12]. Chronic malnutrition in early childhood is strongly associated with poorer cognitive and educational outcomes throughout childhood, lower wages and productivity in adulthood, and lower gross national product on a national level [4]. Global nutrition experts have not yet acknowledged early childhood caries as a contributor to child malnutrition [4], indicating a need for improved communication and collaboration among oral health, nutrition, child health and education professionals. The associations between severe ECC, mouth pain, and malnutrition—particularly at the age of school entry—underscores the serious risk for children’s development and the need to prevent severe ECC to enable children to reach their educational potential. These findings suggest that effective interventions to prevent ECC and its adverse consequences must start within the first 2 years, preferably from pregnancy and birth onward. While we found that access to professional dental care was limited and nearly all ECC were untreated, most mothers received prenatal care and nearly all children received immunizations, indicating that Nepal has an effective infrastructure for maternal–child health services reaching urban and rural populations. This suggests that oral health promotion and dental services (e.g., fluoride varnish, silver diamine fluoride, atraumatic restorative treatment, and extractions) could be more accessible and effective if they were incorporated into general maternal–child health services, as well as early childhood education programs [9,30,31,32].

A strength of this study was the assessment of maternal and child oral health and nutrition knowledge, practices, and health status. In addition, the sizable urban and rural samples yielded statistical power to enable urban–rural comparisons. A limitation of this study was the reliance on a convenience sample that is not representative of the entire population. Moreover, due to time limitations in interviewing mothers, the study inquired about consumption of specific foods and beverages related to dental caries, but did not include a comprehensive food frequency survey. We were also unable to document other acute and chronic health conditions that may have affected children’s oral health and nutrition status and did not conduct an in-depth assessment of children’s oral health-related quality of life. Since behavioral information and mouth pain were self-reported, there may have been reporting biases. Mothers may have under-reported their child’s junk food consumption if they were unaware of what others gave their child or what their child bought on their own. Mothers may have over-reported what they believed was socially acceptable, such as owning a toothbrush. Mouth pain could be over-reported in children under age 3, related to normal teething and non-carious oral infections rather than severe caries, and under-reported in 5–6 year-olds as children with chronic pain may stop complaining about it. In addition, the validity of the dental exams may be limited by calibration procedures among dental examiners. Lastly, although many teachers stated that children with severe caries and mouth pain had poor school attendance and performance, we were unable to collect educational data, and we recommend that future studies collect data on children’s school attendance and performance.

## 5. Conclusions

This study provides evidence that families in urban and rural Nepal are experiencing a nutrition transition in which junk food has become a daily staple of young children’s diet, contributing to high rates of ECC, mouth pain, and malnutrition. As ECC begins in the first 2 years of life, it is critical to prevent ECC by incorporating oral health education and basic dental care into maternal–child health services and early childhood education programs, and to develop policies to prohibit the sale of junk food in and around schools.

## Figures and Tables

**Figure 1 ijerph-16-02456-f001:**
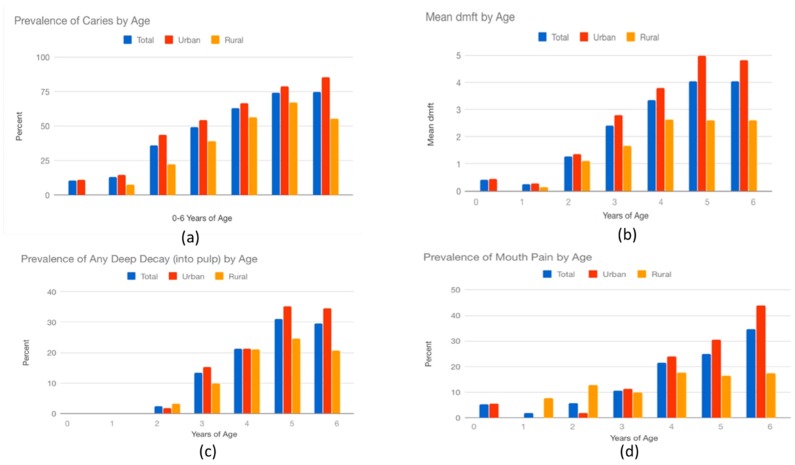
Children’s caries experience by age. (**a**) Prevalence of caries by age; (**b**) mean dmft by age; (**c)** prevalence of any deep decay (into the pulp) by age; (**d**) prevalence of mouth pain by age.

**Table 1 ijerph-16-02456-t001:** Family demographics.

Family Demographics (Sample Size n = 836 Children, 632 Mothers)	Overall Population Mean/Frequency	Urban Population Mean/Frequency	Rural Population Mean/Frequency	Difference *p*-Value ^1^
Child mean age (years)	4.4	4.3	5.1	<0.001
Child Gender (female/male)	52.2% female 46.9% male	53.5% female 44.9% male	51.1% female 48.9% male	NS
Mother mean age (years)	29.7	29.5	29.9	NS
Mother mean education (years)	4.9	6.1	2.5	<0.001
Mean number of children	2.2	2.0	2.7	<0.001
Mean number in household	5.2	5.0	5.7	<0.001
Potable water at home	83.8%	87.9%	75.5%	<0.001
Cooking fuel other than wood (gas)	52.8%	77.7%	1.6%	<0.001
Time to walk from home to a store that sells junk food: Less than 5 min 6–20 min Over 20 min	73.4% 18.0% 8.6%	87.6% 6.1% 6.3%	46.8% 41.6% 11.6%	<0.001 <0.001 <0.001

^1^ Values were obtained with independent samples t-tests. NS = no significance.

**Table 2 ijerph-16-02456-t002:** Oral health and nutrition knowledge and practices.

Family Demographics (*n* = 836 Children, 632 Mothers)	Overall Population Mean/Frequency	Urban Population Mean/Frequency	Rural Population Mean/Frequency	Difference *p*-Value ^1^
**Maternal Knowledge on Caries Risk**				
Eating sweets causes caries	85.5%	84.7%	87.0%	NS
Not brushing causes caries	26.1%	29.5%	20.0%	0.006
Drinking soda/juice causes caries	5.0%	6.4%	2.3%	0.043
Bottle-feeding causes caries	0.2%	0.0%	0.5%	NS
**Maternal Nutrition Practices**				
Daily milk consumption	34.1%	29.5%	42.9%	<0.001
Daily soda consumption	1.9%	2.5%	0.7%	NS
Daily junk food consumption	2.2%	2.9%	0.7%	0.045
**Child Nutrition Practices**				
Breastfed	98.5%	99.0%	97.4%	NS
Mean duration of breastfeeding (months)	28.6	27.3	30.9	<0.001
Bottle-fed	16.8%	22.2%	7.9%	<0.001
Mean duration of bottle-feeding (months)	15.8	16.1	14.5	<0.001
Bottle-fed with sugary drink	9.6%	10.7%	4.3%	NS
Bottle during sleep (occasionally/frequently)	33.6%	27.3%	63.6%	<0.001
Daily consumption of milk by age Total 0–2 years 3–6 years	68.2% 73.0% 66.9%	65.9% 75.4% 63.1%	72.5% 66.7% 73.7%	NS NS <0.001
Daily consumption of soda/juice by age Total 0–2 years 3–6 years	1.3% 0.6% 1.4%	1.4% 0.8% 1.5%	1.1% 0.0% 1.3%	NS NS NS
Daily consumption of sweets by age Total 0–2 years 3–6 years	50.8% 34.4% 55.1%	62.0% 36.4% 69.6%	29.7% 28.9% 29.8%	<0.001 NS <0.001
Daily consumption of junk food by age Total 0–2 years 3–6 years	24.0% 12.9% 26.8%	31.2% 14.4% 36.2%	10.3% 8.9% 10.5%	<0.001 NS <0.001
**Maternal Oral Health Practices**				
Has her own toothbrush	92.8%	95.5%	86.4%	<0.001
Has been to the dentist	50.3%	53.2%	43.1%	<0.001
Received prenatal care; mean number of visits	80.6%; 5.3	86.3%; 5.9	70.3%; 4.0	<0.001
**Child Oral Health Practices**				
Has his/her own toothbrush	74.4%	84.4%	56.0%	<0.001
Has toothpaste	90.4%	93.7%	84.2%	<0.001
Mother helps with brushing frequently/almost always	21.9%	25.6%	15.5%	<0.001
Mother does nothing to care for child’s teeth	14.1%	10.6%	20.6%	<0.001
Has been to the dentist	10.3%	12.4%	6.3%	0.007
Up-to-date immunizations	97.4%	96.2%	99.6%	<0.001

^1^ Values were obtained with independent samples t-tests. NS = no significance.

**Table 3 ijerph-16-02456-t003:** Children’s oral health and nutrition status.

	Overall Population Mean/Frequency	Urban Population Mean/Frequency	Rural Population Mean/Frequency	Difference *p*-Value ^1^
**Oral Health Status**				
Prevalence of caries	58.2%	62.2%	51.2%	0.002
Mean proportion of untreated caries (decayed teeth/ decayed, missing, or filled teeth = d/dmft)	93.6%	94.8%	90.9%	NS
Range in number of dmft	0–19	0–17	0–19	NS
Mean number of dmft for all children	2.9	3.3	2.1	<0.001
Mean number of dmft for children with caries	4.3	5.3	2.7	<0.001
Distribution of number of decayed teeth 1–4 5–9 >10	50.1% 35.3% 14.6%	51.6% 34.1% 14.4%	47.8% 37.3% 14.8%	NS NS NS
Prevalence of deep decay into the pulp	19.6%	21.1%	16.8%	NS
Prevalence of mouth pain Any mouth pain (occasionally/frequently/always) Mouth pain frequently/always	20.0% 1.9%	22.7% 2.1%	15.0% 1.5%	<0.001 <0.001
Mother’s assessment of child’s oral health as “bad”	19.7%	20.8%	25.0%	<0.001
Mother’s assessment of child’s overall health as “bad”	4.4%	6.0%	1.5%	0.006
**Nutrition Status**				
Mean Z-score Height-for-Age (HAZ) < −2 (stunting)	34.2%	29.5%	42.7%	<0.001
Mean Z-score Weight-for-Age (WAZ) < −2 (underweight)	14.8%	13.2%	17.6%	NS
Mean Z-score BMI-for-Age (BAZ) < −2 (wasting)	2.1%	2.1%	2.0%	NS

^1^ Values were obtained with independent samples t-tests. NS = no significance.

**Table 4 ijerph-16-02456-t004:** Associations between oral health and nutrition.

**Predictive Variable**	**Outcome Variable**	**Correlation Coefficient** **B**	**Odds Ratio Exp (B)/F (df)**	**Correlation Factor** **R Square**	***p*-Value ^1^**
**Mother’s Frequent Consumption of** milk soda/juice junk food	**Child’s Frequent Consumption of** milk soda/juice junk food	2.472 −16.640 2.032	11.848 0.000 7.633	0.243 0.003 0.031	<0.001 no significance <0.001
**Child’s Frequent Consumption of** soda/juice sweets junk food	**Caries Severity (dmft)**	1.778 +1.05 (soda) +0.21 (sweets) −0.20 (junk food)	5.391 (3, 329)	0.038	<0.001
**Caries Severity (dmft)**	**Any mouth pain**	0.225	1.253	0.188	<0.001

**Predictive Variable**	**Outcome Variable**	**Estimate**	**Standard Error**	***p*-Value**	**n ^2^**
**Caries Severity (dmft)**	**Overall Population** **Weight-for-Age (WAZ) < −2** **Height-for-Age (HAZ) < −2** **BMI-for-Age (BAZ) < −2**	−0.33 −0.30 −0.36	0.06 0.04 0.16	<0.0001 <0.0001 0.0298	836 836 836
**Caries (Deep Decay into the Pulp)**	**Overall Population** **WAZ < −2** **HAZ < −2** **BAZ < −2** **Urban Population** **WAZ < −2** **HAZ < −2** **BAZ < −2** **Rural Population** **WAZ < −2** **HAZ < −2** **BAZ < −2**	−0.57 −0.74 0.22 −0.3 −0.46 −0.2 −1.06 −1.1839 0.89	0.28 0.21 0.59 0.33 0.25 0.8 0.54 0.36 0.88	0.04 <0.001 NS NS NS NS 0.05 <0.001 NS	122 281 17 70 155 11 52 126 6
**Any mouth pain**	**Overall Population** **WAZ < −2** **HAZ < −2** **BAZ < −2**	−0.0154 −0.4205 0.17	0.11 0.19 0.58	NS 0.03 NS	122 281 17

^1^ Values were obtained with linear and multivariate linear regressions; ^2^ values were obtained with binomial logistic regressions. NS = no significance.

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
