# Peer review of "Early Childhood Oral Health and Nutrition in Urban and Rural Nepal"

_ijerph, 2019, doi:10.3390/ijerph16142456_

Round 1

Reviewer 1 Report

Dear authors, this study is well conducted and its epidemiological results are important for rural communities around the world.

I suggest to:

modify keyword according to mesh (MEdical Subject Headings) word: https://meshb.nlm.nih.gov/search

M&M and results section are well drafted.

No information about QoL and patients general health condition have been provided, so please shortly refer on how some disease or syndromes could influence oral health:

10.3390/biomedicines7020033; 

10.1155/2019/5907195

10.1155/2018/7848735

10.23736/S0926-4970.16.03995-3

Thank You

Best regards

Reviewer 2 Report

This was a well planned and significant study.  It is well written and provides pertinent background information. I would have liked to see stronger data about "stunting, and its implication for the country of Nepal.  It would also be interesting to know whether any of the interventions you mentioned, such a better advertisement, be implemented? 

Stunting, in relationship to a healthy work force and gross national product, might be a major incentive for the government to intervene. 

**Please consider adding data and discuss regarding stunting.

I approve your article and would like to see more study's of this magnitude..
